# Unveiling the Dominant Control of the Systematic Cooling Bias in CMIP6 Models: Quantification and Corrective Strategies

Jie Zhang[1,2*], Kalli Furtado[3*], Steven T. Turnock[4,5], Yixiong Lu[1,2], Tongwen Wu[1,2],

Fang Zhang[1,2], Xiaoge Xin[1,2], Yuyun Liu[6]

1. *State Key Laboratory of Disaster Weather Science and Technology, CMA EMPC*
2. *CMA Earth System Modeling and Prediction Centre, China Meteorological Administration,Beijing, China*
3. *National Oceanography Center, Southampton, UK*
4. *Met Office Hadley Centre, Exeter, UK*
5. *University of Leeds Met Office Strategic (LUMOS) Research Group, University of Leeds, Leeds, UK*
6. *Center for Monsoon System Research, Institute of Atmospheric Physics, Chinese Academy of Sciences, Beijing, China*

*Corresponding to: Jie ZHANG (jiezhang@cma.gov.cn) & Kalli Furtado (kalli.furtado@noc.ac.uk)*

## Abstract

Including sophisticated aerosol schemes in the models of the sixth Coupled Model Inter-comparison Project (CMIP6) has not improved historical climate simulations. In particular, the models underestimate the surface air temperature anomaly (SATa) when anthropogenic sulfur emissions increased in 1960-1990, making the reliability of the CMIP6 projections questionable. This cooling bias is largely attributable to the unreasonable simulated atmospheric sulfate burden changes. Sulfate burden anomaly are closely linked to both sulfate and $SO_2$ deposition processes. Intensified sulfate deposition directly reduces atmospheric sulfate loading, while enhanced $SO_2$ deposition limits precursor availability for sulfate formation by oxidation. These deposition processes regulate sulfate concentrations directly and indirectly. The systematically underestimated sulfate turnover time in CMIP6 models suggests that refining $SO_2$ deposition process rather than sulfate deposition would be a more scientific approach for model improvement. This is supported by two post-CMIP6 models that show better SATa reproduction after improving the $SO_2$ deposition parameterizations. Strong correlations between sulfate burden anomaly and SATa persist before, during, and after the 1960-1990 period. Such temporal consistency confirms the dominant role of sulfate-related physical processes across all examined time intervals.

## 1. Introduction

Atmospheric aerosols have rapidly increased since the Industrial Revolution. Over this time period, the total aerosol effective radiative forcing (ERF) was dominated by the sulfate cooling effect, which offsets a substantial portion of global-mean forcing from well-mixed greenhouse gases (IPCC, 2023). Without this historical aerosol ERF, the Paris Agreement's target of limiting global warming to 1.5°C above pre-industrial levels would have already been missed in 2015 (Hienola et al., 2018). Similarly, stopping all present-day anthropogenic aerosol emissions is estimated to induce a global-mean surface heating of 0.5-1.1°C (Samset et al., 2018). The year 2024 has been confirmed as the hottest year in human history and was the first year to breach the 1.5°C warming limit (Bevacqua et al., 2025). Moreover, recent accelerated temperature trends may be attributable to reductions in atmospheric aerosols, particularly from reduced commercial shipping emissions. Hansen et al. (2025) suggest that even small emissions in relatively pristine air have substantial effects, highlighting the crucial need to improve the representation of aerosol effects in global climate models for more reliable projections.

The observed temporal evolution of historical surface air temperature (SAT) is one of the major metrics used for evaluating the performance of climate models. However, the SAT anomalies (SATa) in the CMIP6 models are systematically lower than observations during the 1960-1990 period, whereas the CMIP5 models, on average, track the instrumental record quite well (e.g., Flynn and Mauritsen, 2020). The 1960-1990 period, when the cooling bias prevailed, is coincident with the so-called Great Acceleration period, during which human activities intensified remarkably and led to global-scale impacts on the Earth System (Steffen et al., 2007). Recent studies hypothesized that aerosol forcing in CMIP6 is stronger than in CMIP5 and is responsible for the suppressed late 20[th]-century warming (e.g., Dittus et al., 2020; Smith and Forster, 2021).

Given that all CMIP6 models use identical anthropogenic $SO_2$ emissions (Hoesly et al., 2018), the cooling anomaly points towards a problem with the sulfur cycle in recent

earth system models or the emissions data (Hardacre et al., 2021; Wang et al., 2021).
In this study, we examine the sulfate-related processes in eleven CMIP6 models with
aerosol schemes. We will identify the key processes governing sulfate burden in these
models and provide recommendations for further model improvements.

**2.  Model, data, and method**
**2.1 CMIP6 models and data**
**Table 1.** Information of the eleven CMIP6 models with aerosol schemes.

| Model | Country | Interactive Chemistry | Members | Reference |
|---|---|---|---|---|
| **BCC-ESM1** | China | Yes | 3 | Wu et al., (2020); Zhang et al., (2021b) |
| **CESM2** | US | No | 11 | Danabasoglu et al. (2020) |
| **CESM2-FV2** | US | No | 3 | Danabasoglu et al. (2020) |
| **EC-Earth3-AerChem** | European consortium | Yes | 2 | Döscher et al. (2021) |
| **GFDL-ESM4** | US | Yes | 3 | Dunne et al. (2020) |
| **MIROC6** | Japan | No | 50 | Tatebe et al. (2019) |
| **MIROC-ES2L** | Japan | No | 30 | Hajima et al. (2020) |
| **MPI-ESM-1-2-HAM** | Germany | Yes | 3 | Mauritsen et al. (2019) |
| **MRI-ESM2-0** | Japan | Yes | 10 | Yukimoto et al. (2019) |
| **NorESM2-LM** | Norway | Yes | 3 | Seland et al. (2020) |
| **UKESM1-0-LL** | UK | Yes | 19 | Sellar et al. (2019) |


Eleven CMIP6 climate models with interactive aerosol schemes are employed in
this study, including seven models with interactive chemistry and four without (Table
1). The outputs from two CMIP6 experiments are used: (1) the historical experiment,
which simulates climate evolution from 1850 to 2014, forced by time-varying external
forcings from natural processes (e.g., solar activity, volcanic eruptions) and
anthropogenic factors (e.g., greenhouse gas, aerosol emissions, land-use changes). All
the available realizations for each model were used to minimize the uncertainty from
internal variability in the climate system; (2) the 1pctCO2 simulations, in which $CO_2$ is
gradually increased at a rate of 1% per year. The 1pctCO2 experiment is designed for
studying model responses to $CO_2$ and is somewhat more realistic than rapidly
increasing $CO_2$, such as in the abrupt-4×CO2 experiment. Historical experiment outputs
from two post-CMIP6 models, BCC-ESM1-1 and UKESM1-1-LL, with revised $SO_2$
deposition parameterizations are also included in this study.
The model outputs used in this study include SAT and eight key sulfur-cycle
variables: sulfate aerosol concentration, sulfate wet and dry deposition rates, sulfur
dioxide concentration ($SO_2$), $SO_2$ wet and dry deposition rates, gas-phase and aqueous-
phase oxidations of $SO_2$ to sulfate particles. For these sulfur-cycle variables, the inter-
member variability within the historical experiment is substantially smaller than that of
SAT. For instance, across the 11 CESM2 members, the standard deviation of sulfate
burden is only about 4% of its interannual variability during 1960-1990, whereas the
corresponding value for SAT is approximately 21%. Similar results are also evident in
the 19 UKESM1 members, where the standard deviation of sulfate burden is 3% of its
interannual variability, compared to 32% for SAT. Given that inter-member variability
in sulfur-cycle variables is relatively small relative to their interannual fluctuations, we
therefore use the first realization of the historical simulations and neglect inter-member
differences for these sulfur-cycle variables.
Monthly mean SAT from the Met Office Hadley Centre/Climatic Research Unit
global surface temperature dataset version 5 (HadCRUT5) from 1850 to 2014 are used
for model evaluations (Morice et al., 2021). Considering the scarcity of long-term

reliable observations in polar regions, we focus on SAT changes within the latitudinal belt from 60ºS to 65ºN. The 'global' mean SAT is calculated as the area-weighted average over this latitudinal belt.

## 2.2 SO₂ turnover time and sulfate turnover time

Atmospheric sulfate concentrations are governed by the emission and oxidation of its precursors, as well as deposition processes. Anthropogenic SO₂ emissions are the major source of sulfate aerosol over land in polluted regions. Given that CMIP6 models typically employ identical anthropogenic SO₂ emission inventories, the inter-model spread in simulated sulfate concentrations primarily stems from discrepancies in SO₂-to-sulfate oxidation rates and sulfate deposition velocities. Here we define the atmospheric residence time of SO₂ and sulfate aerosols as follows.

SO₂ turnover time is determined by its atmospheric burden and its total loss rate, which includes both deposition and chemical oxidation to sulfate. It is defined as:

$$\tau_{SO_2} = \frac{B_{SO_2}}{(R_{dSO_2} + R_{oSO_2})} \quad (1),$$

where $\tau_{SO_2}$ is the SO₂ turnover time, $B_{SO_2}$ is the global mean atmospheric SO₂ burden, $R_{dSO_2}$ is the total SO₂ deposition rate including both wet and dry depositions, and $R_{oSO_2}$ is the oxidation rate of SO₂ to sulfate via gas-phase and aqueous-phase chemistry.

Sulfate turnover time is defined as:

$$\tau_{SO_4} = \frac{B_{SO_4}}{R_{dSO_4}} \quad (2),$$

where $\tau_{SO_4}$ is the sulfate turnover time, $B_{SO_4}$ is the global mean atmospheric sulfate burden, and $R_{dSO_4}$ is the global mean total sulfate deposition rate including both wet and dry depositions.

## 2.3 The transient Climate Response (TCR) index

The Transient Climate Response (TCR) index is calculated as the mean SAT anomaly over a 20-year period centered on the year when atmospheric $CO_2$ concentration has doubled in the 1pctCO2 simulation. It is an important metric for quantifying $CO_2$-induced historical warming and has been widely used for model evaluations and intercomparison studies (e.g., Bevacqua et al., 2025; O'neill et al., 2016).

## 3. Results

### 3.1 SATa and sulfate burden anomaly

The historical evolutions of global mean SATa in the eleven CMIP6 models with interactive aerosol schemes are shown in Fig. 1a. All the models tend to underestimate SATa since the 1930s. The cooling anomaly in the CMIP6 model marked a notable departure from earlier model generations, which can effectively capture the instrumental SAT record with observations falling well within model spread (e.g., Flynn and Mauritsen, 2020; Hegerl, et al., 2007).

The cooling bias is most pronounced from 1960 to 1990. The SATa is about 0.34$^{\rm o}$C in the observations. However, the multi-model mean (MMM) SATa is about 0.3$^{\rm o}$C lower with a large model spread. The SATa ranges from -0.24$^{\rm o}$C in EC-Earth3-AerChem to 0.19$^{\rm o}$C in GFDL-ESM4 and MIROC6. The cooling is noticeable at the mid to high latitude in the Northern Hemisphere (as shown in the attached SATa map in Fig.1a). The sudden drop in SATa in the early 1960s and 1990s may be due to the stronger model responses to large volcanic eruptions, Mount Agung in 1963 and Mount Pinatubo in 1991, than in the observations (Chylek et al., 2020). The cooling biases diminish in later periods, corresponding to the generally high model sensitivity to greenhouse gas forcing (Smith and Forster, 2021).

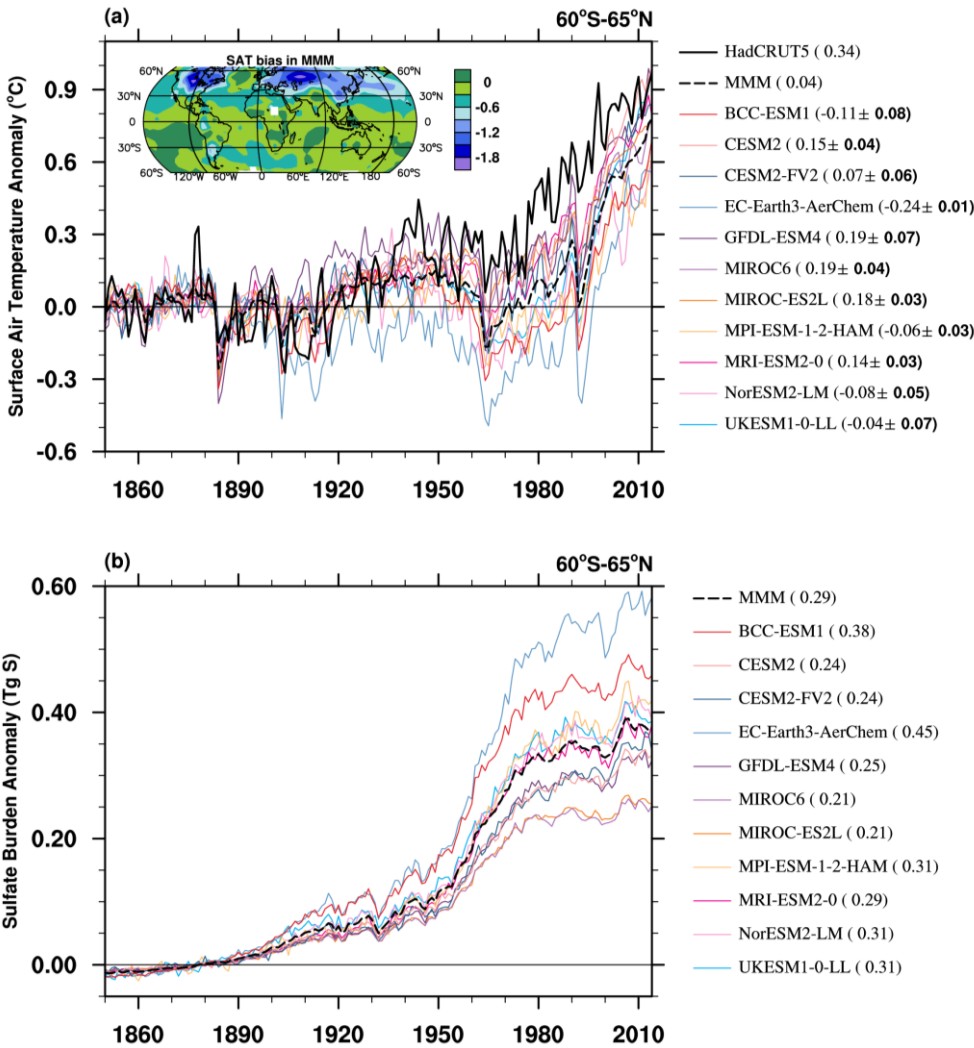

**Figure 1.** (a) Historical surface air temperature anomalies (SATa) relative to 1850-1900 mean from HadCRUT5 (thick black line), the ensemble mean of each CMIP6 model (solid colored lines), and the multi-model mean (MMM; dashed black line). Numbers in parentheses indicate the mean SATa for each model during 1960-1990, with the inter-member spread shown as ± one standard deviation. Units: °C. (b) Same as (a), but for sulfate burden anomalies for the first realization of each CMIP6 model (colored lines) and the MMM (dashed black line). Units: Tg S.

The cooling bias in CMIP6 models coincides with the rapid increase in anthropogenic emissions, particularly of $SO_2$, the primary precursor of atmospheric sulfate (Zhang et al., 2021a). Global $SO_2$ emissions grew steadily after the 1950s and peaked in the 1970s at approximately 180Tg yr$^{-1}$, about 3.6 times the level of the 1950s (Hoesly et al., 2018). The rise in $SO_2$ emissions has directly contributed to elevated sulfate concentrations in the troposphere. The temporal evolution of sulfate burden shows a significant upward trend aligned with the anthropogenic emission (Fig.1b),

initially driven by industrialization and further accelerated after the 1950s mainly due to intensified anthropogenic $SO_2$ emission from industries and the energy-transformation sectors (e.g., Ohara et al., 2007; Vestreng et al., 2007). The increased sulfate burden interrupted a decades-long warming trend through the cooling effect of sulfate aerosols, even as atmospheric $CO_2$ concentrations continued to rise (Wilcox et al., 2013).

Due to emission-control policies implemented in Europe and North America (Aas et al., 2019; Hand et al., 2012; Vestreng et al., 2007), such as the Gothenburg Protocol (Eb, 1999) and the 1990 Clean Air Act Amendments in the U.S. (Likens et al., 2001), global anthropogenic $SO_2$ emissions were suppressed after the 1980s and SAT started to rise rapidly in both observation and model simulations. It should be noted that the CMIP6 emission inventory does not fully capture the early 21$^{st}$ century $SO_2$ emission reductions in East Asia (Wang et al., 2021). However, this period lies outside the 1960-1990 focus of the present study, and its impact on SAT reproduction is beyond the main scope of this paper.

The systematically underestimated SATa suggests an excessively strong sulfate-induced cooling effect in CMIP6 models, as indicated by the contrasting performance of individual models. For instance, the MIROC models exhibit the lowest sulfate burden (0.21 Tg S) and smallest cooling bias relative to observation (0.15°C below HadCRUT5) during 1960-1990, while EC-Earth3-AerChem generates a sulfate burden approximately double that value (0.45 Tg S) and nearly four times the cooling bias (0.58°C below HadCRUT5). Analysis across the 11 CMIP6 models reveals a statistically significant negative correlation of -0.92 between sulfate burden anomalies and SATa (Fig. 2a). This relationship highlights the potential role of overestimated sulfate-induced cooling in driving the inter-model spread of SATa biases.

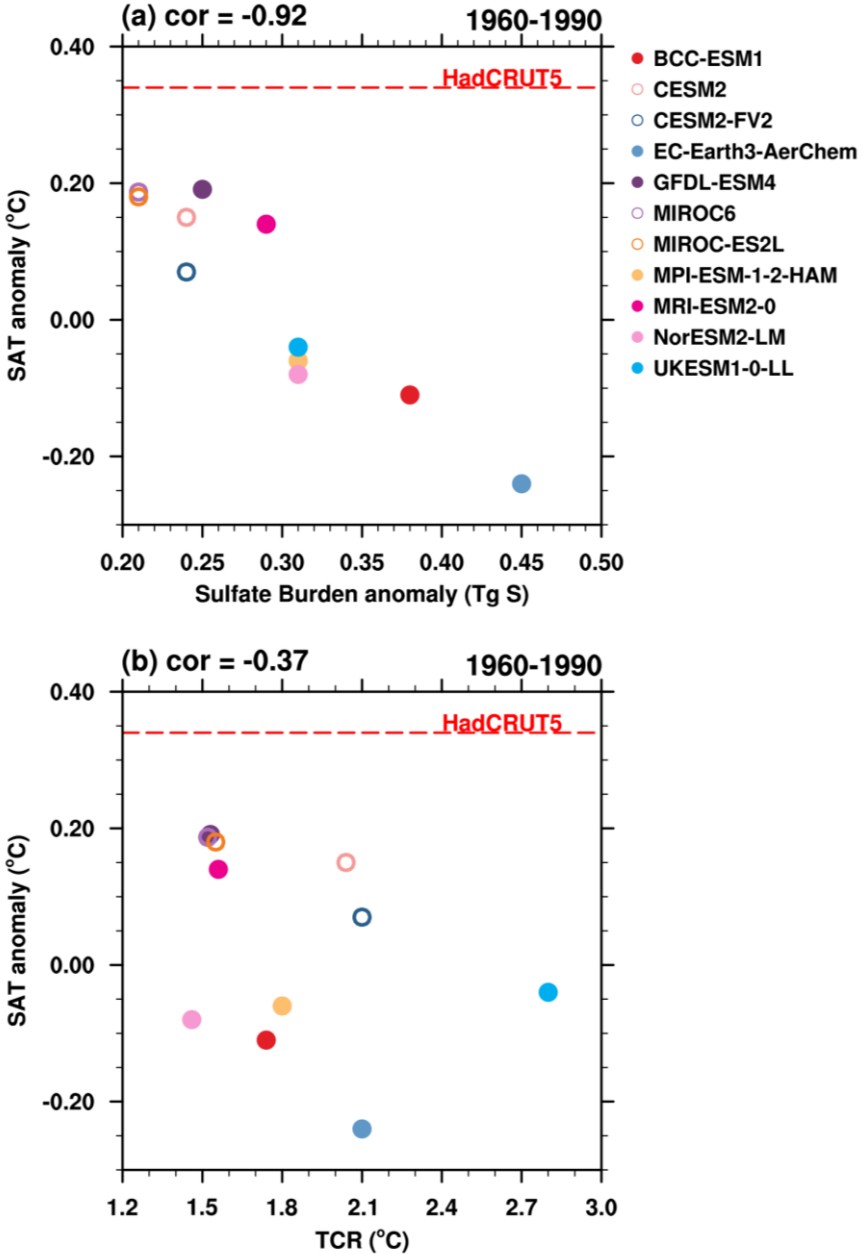

**Figure 2.** (a) Scatter plots of sulfate burden anomaly versus SATa, and (b) scatter plot of TCR versus SATa during 1960-1990 from historical experiments. Anomalies are calculated relative to the 1850-1900 mean. Models with and without interactive chemistry are denoted by colored dots and colored circles, respectively. The corresponding correlation coefficient (cor) for each panel is shown in the upper-left corner. The red dashed line refers to SATa in HadCRUT5.

Interactive chemistry may affect sulfate formation and sulfate aerosol burdens in the atmosphere (Mulcahy et al., 2020). Models with interactive chemistry (colored dots in Fig.2a) generally show higher sulfate burdens and lower SATa than non-interactive models (colored circles). However, the relationship between sulfate burden anomaly

and SATa is a robust feature across CMIP6 models, independent of their chemical complexity.

Greenhouse gases (GHGs) also increased rapidly during 1960-1990. However, TCR, which can generally indicate the impact of GHGs, is insignificantly correlated with SATa in CMIP6 models, and the correlation coefficient across models is even negative (Fig.2b). Therefore, the inter-model spread in cooling biases can substantially be attributed to discrepancies in simulated sulfate aerosol burden.

It should be noticed that there are fast and slow components of global warming in response to radiative forcing changes (Held et al., 2010). The fast component, characterized by an exponential decay timescale of less than 5 years, is primarily driven by rapid adjustments in the upper ocean layers. In contrast, the slow component evolves over centuries and is associated with heat uptake by deeper ocean layers. Lagged oceanic and dynamical feedbacks will further delay and modulate warming rates (Chen et al., 2016; Watterson and Dix, 2005). In this study, the fast response to sulfate forcing can be rapidly detected by SATa, especially when the sulfate forcing is sustained during 1960-1990. Moreover, the global mean perspective in this study makes the results insensitive to the impact of spatial redistribution of temperature anomalies caused by dynamical feedbacks.

**3.2 Sulfur Deposition rates and $SO_2$ oxidation rate**

$SO_2$ deposition, sulfate deposition, and $SO_2$ oxidation to sulfate are the key processes governing the atmospheric sulfur cycle. About half of the $SO_2$ emission is removed by dry deposition at the surface and through wet scavenging by precipitation (e.g., Chin et al., 1996). The remaining fraction is oxidized to sulfate, mainly through two pathways: gas-phase reaction with the hydroxyl radical (OH), and aqueous-phase oxidation within cloud and fog droplets, where reactions with ozone ($O_3$) and hydrogen peroxide ($H_2O_2$) are dominant. These processes are critical determinants of atmospheric sulfate burden.

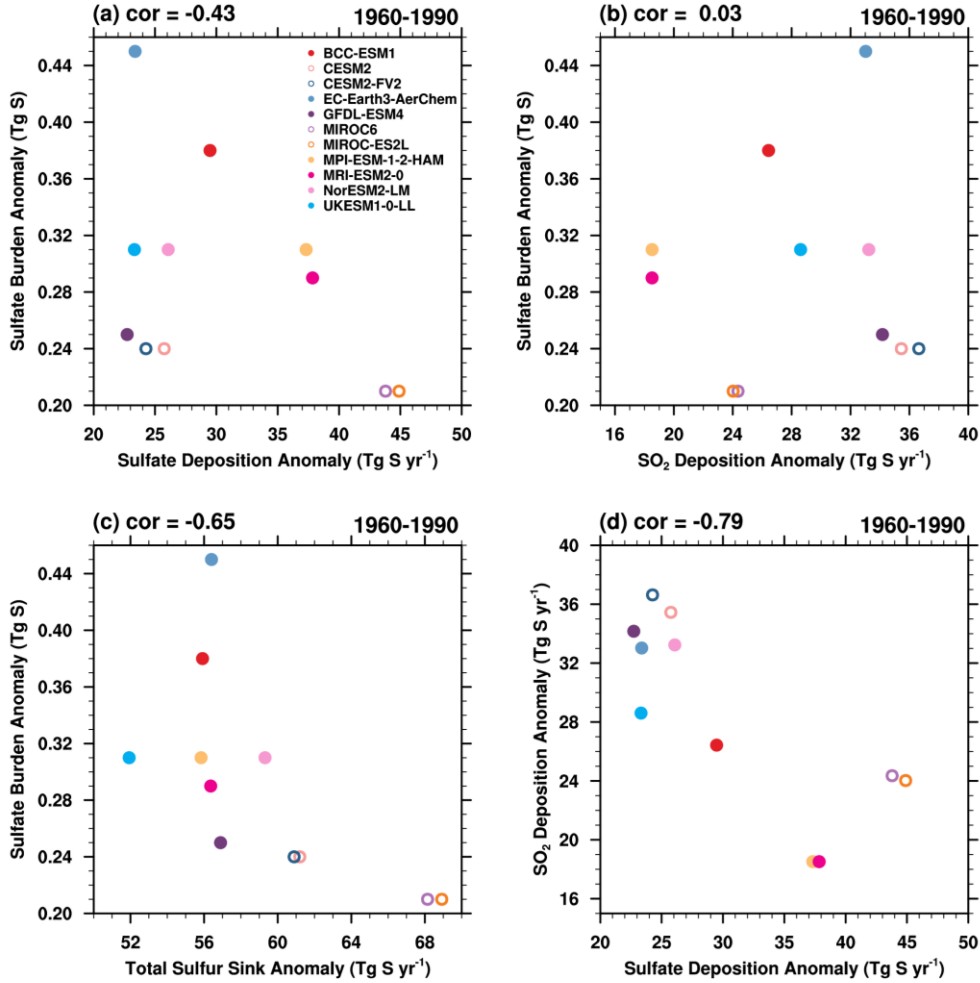

**Figure 3.** (a) Sulfate deposition anomaly, (b) SO₂ deposition anomaly, and (c) total sulfur sink anomaly (x-axis) versus sulfate burden anomaly (Tg S, y-axis) in each model during 1960-1990. (d) Sulfate deposition anomaly (x-axis) versus SO₂ deposition anomaly (y-axis) during 1960-1990. Units for deposition anomalies are Tg S yr⁻¹.

Fig. 3 shows the inter-model relationship between global mean anomalies of sulfate burdens and sulfur depositions during 1960-1990, relative to the pre-industrial baseline (1850-1900). The sulfate burden anomaly is negatively correlated with sulfate deposition anomaly. However, the correlation is statistically insignificant. This may be partly attributable to a subset of five models characterized by both low sulfate burden and low sulfate deposition anomalies. These models degrade the robustness of the linear fit derived from the remaining models. There is no clear statistical relationship between sulfate burden anomaly and SO₂ deposition anomaly (Fig. 3b). However, when considering the total sulfur sink anomaly, including both sulfate and SO₂ deposition

anomalies, the correlation with sulfate burden anomaly strengthens to -0.65, significant
at the 5% level using a Student's t-test (Fig.3c). Notably, within the subset of five
models, most show higher $SO_2$ deposition anomaly in relative to the multi-model mean.
This high $SO_2$ deposition anomaly compensates for their low sulfate deposition
anomaly, influencing the total sulfur deposition magnitude sufficiently to sustain a
significant correlation with sulfate burden anomaly in these models. Further analysis
reveals a strong negative correlation (-0.79) between $SO_2$ deposition rate anomaly and
sulfate deposition rate anomaly, suggesting a compensatory relationship between these
two sulfur removal pathways (Fig.3d).

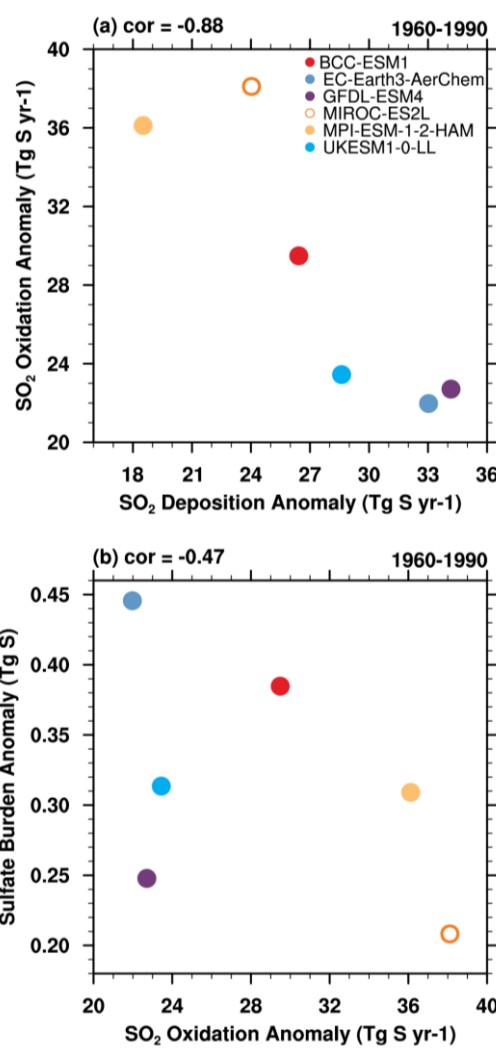


**Figure 4. (a)** $SO_2$ deposition anomaly versus $SO_2$ oxidation anomaly, and **(b)** $SO_2$ oxidation
anomaly versus sulfate burden anomaly in each model during 1960-1990.
The formation of atmospheric sulfate aerosol is governed by the balance between
the loss of its precursor, $SO_2$, and its chemical transformation. As shown in Fig.4a,
inter-model comparisons show a significant anti-correlation between $SO_2$ deposition
anomaly and the oxidation rate anomaly across the six models for which relevant data
are available for calculation (-0.88). That is, enhanced $SO_2$ deposition rate, particularly
through dry deposition processes, limits the availability of $SO_2$ for oxidation to sulfate.
The relationship between oxidation rate anomalies and the sulfate burden anomalies is
negative but not statistically robust within this limited model subset. A more
comprehensive analysis with a larger model ensemble is needed to robustly quantify
the relative contributions of oxidation pathways to the sulfate aerosol burden.
Therefore, biases in sulfate burden simulations arise either directly from sulfate
deposition or indirectly from $SO_2$ deposition, which limits the availability of $SO_2$ for
oxidation.

**3.3 $SO_2$ turnover time and sulfate turnover time**
$SO_2$ deposition, sulfate deposition, and $SO_2$ oxidation rate determine the respective
turnover times for $SO_2$ and sulfate, which quantify their mean atmospheric residence
times before removal. Here we examine $SO_2$ turnover time and sulfate turnover time,
quantities with clear physical interpretations, to identify the dominant physical and
chemical processes responsible for the sulfate burden biases.

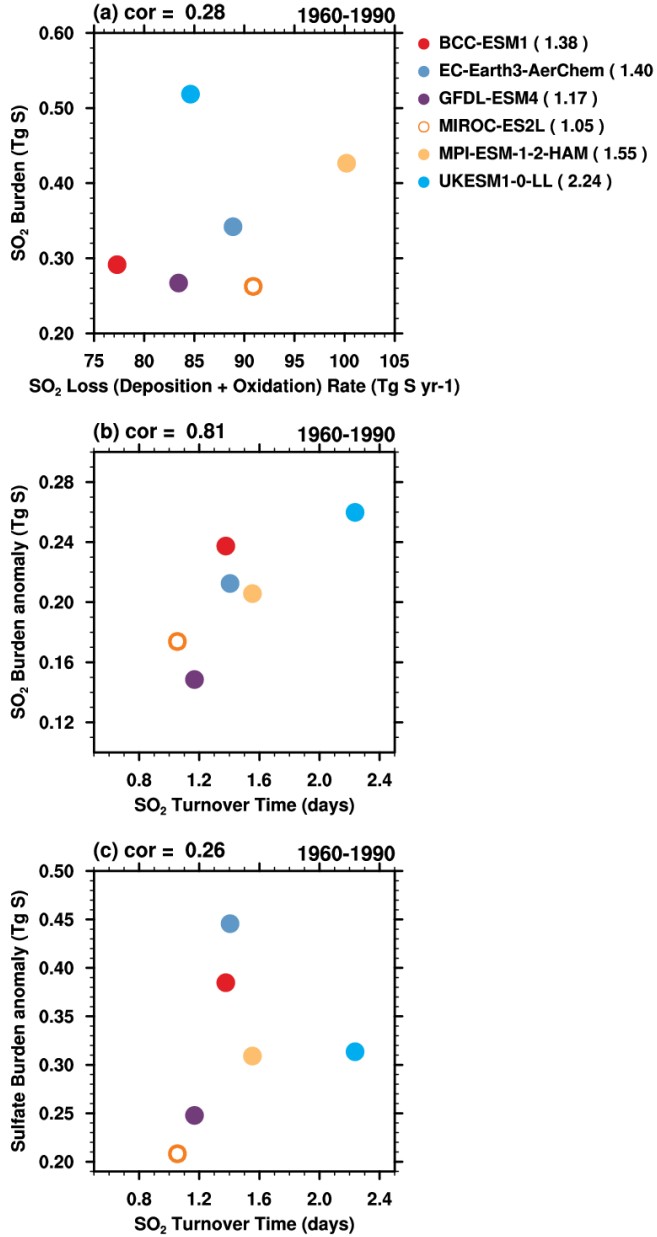


**Figure 5.** (a) $SO_2$ loss rate versus $SO_2$ burden in 1960-1990. $SO_2$ loss rate includes $SO_2$ deposition and oxidation. (b) $SO_2$ turnover time versus $SO_2$ burden anomaly in 1960-1990. (c) $SO_2$ turnover time versus sulfate burden anomaly in 1960-1990.

The correlations between $SO_2$ burden and its total loss rate, including both deposition and chemical oxidation, are notably weak (Fig.5a). Given that the models share identical anthropogenic $SO_2$ emission inventories, this poor correlation likely stems from substantial inter-model differences in the representation of natural $SO_2$ precursor emissions (e.g., from oceanic dimethyl sulfide) and their subsequent

atmospheric processing. The SO₂ turnover time ($\tau_{SO_2}$) as defined in Eq. 1, ranges from
1.05 to 2.24 days in the CMIP6 models. The $\tau_{SO_2}$ is highly correlated with SO₂ burden
anomaly with a correlation coefficient of 0.81 (Fig.5b). However, its correlation with
the sulfate burden anomaly is weak (Fig.5c).

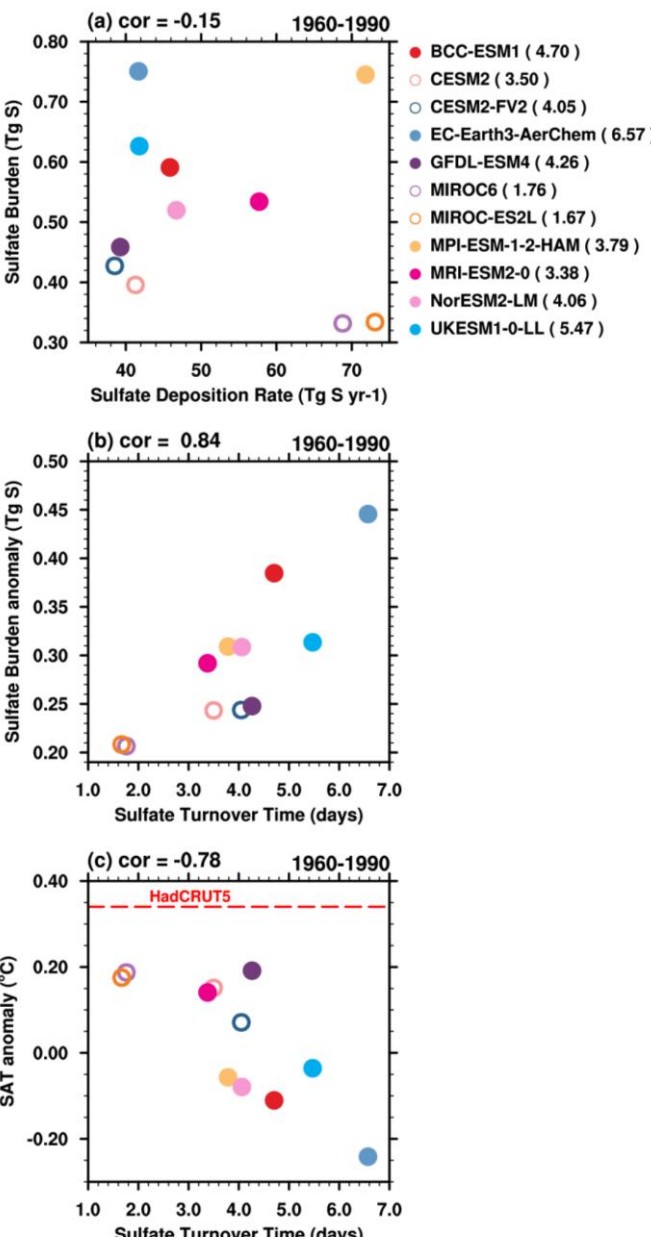


**Figure 6.** (a) Sulfate deposition rate versus sulfate burden during 1960-1990. (b) Sulfate turnover
time versus sulfate burden anomaly during 1960-1990. (c) Sulfate turnover time versus SATa during
1960-1990. The red dashed line refers to SATa in HadCRUT5.

293       Figure 6 presents the simulated sulfate deposition and sulfate burden in 1960-1990.

The weak negative correlation (-0.15) indicates that sulfate deposition alone cannot
fully explain inter-model differences in sulfate burden. Sulfate turnover time is
quantified following Eq. (2) in Section 2.2 as the ratio of sulfate burden to sulfate
deposition, representing the average atmospheric residence time of sulfate aerosols.
The sulfate turnover time exhibits considerable inter-model variability, ranging from
1.67 days in MIROC-ES2L to 6.57 days in EC-Earth3-AerChem. These results
generally agree with most aerosol models, which typically simulate sulfate lifetimes
of around 4 days (e.g., Textor et al.,2006; Liu et al., 2012; Matsui and Mahowald,
2017; Tegen et al., 2019). However, sulfate turnover times in models are notably
shorter than observational estimates, such as 7.3 days (0.02 yr) in Charlson et al.
(1992) and 10-14 days in Kristiansen et al. (2012). This discrepancy may stem from
premature removal processes, inadequate poleward transport, or incomplete chemical
representations (e.g., Croft et al., 2014).
The inter-model variations in sulfate turnover time exhibit a strong correlation with
sulfate burden anomalies and SATa during the 1960-1990 period, with a correlation
coefficient of 0.84 and -0.78 (Fig.6b and Fig.6c). This suggests that differences in
sulfate turnover time may account for both the sulfate burden anomaly variations and
the consequent surface temperature differences among models. CMIP6 models
systematically overestimate sulfate burden anomalies, implying that these models
should exhibit shorter lifetimes to produce lower sulfate burden anomalies and higher
SATa (Fig.6c). However, enhancing sulfate deposition to reduce burden anomalies is
not a physically reasonable solution, as it would worsen the already too-short
simulated sulfate aerosol lifetime.
Therefore, as indicated by section 3.2, model improvement efforts should
prioritize $SO_2$ deposition process refinement rather than sulfate deposition adjustment
as a more scientifically sound approach.

**3.4 The performances in the two post-CMIP6 models**

To suppress the substantial cold bias in the BCC-ESM1 model, which underestimates the observed SATa by 0.45°C during the 1960-1990 period, we increase the dry deposition velocity of $SO_2$ by a factor of four over land surface and by a factor of 1.5 over the ocean to reduce the availability of $SO_2$ for oxidation. This effect is similar to that in UKESM1-0-LL by improving $SO_2$ dry deposition parameterization (Hardacre et al., 2021; Mulcahy et al., 2023). The impact of changes to the $SO_2$ dry deposition parameterization in UKESM1-0-LL is an increase of $SO_2$ dry deposition by a factor of 2 to 4. Accordingly, SATa increases to 0.45°C in BCC-ESM1-1 and rises to 0.25°C in UKESM1-LL. Sulfate turnover time in the two post-CMIP6 models, 8.53 days in BCC-ESM1-1 and 5.77 days in UKESM1-1-LL, is generally longer than that of their CMIP6 versions. The longer sulfate lifetimes in the two post-CMIP6 models may be due to lower $SO_2$ in these revised models, but also could be due to physical climate changes (e.g., temperatures, clouds, rainfall).

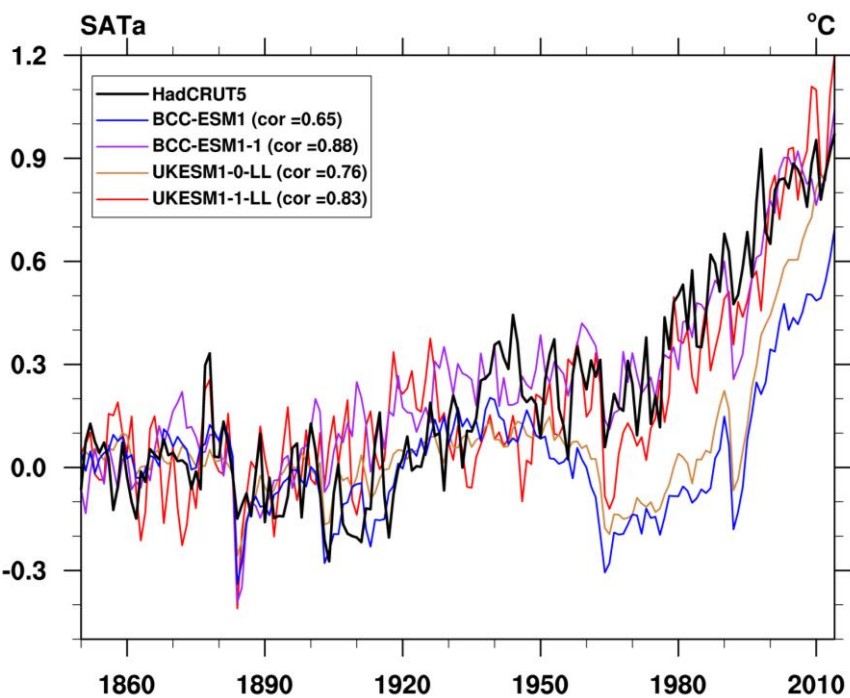

**Figure 7.** Evolutions of SATa relative to 1850-1900 mean for HadCRUT5, BCC-ESM models, and UKESM models. The numbers in legend are the corresponding correlation coefficients with HadCRUT5.

As demonstrated by the global mean SATa in BCC-ESM1-1 and UKESM1-1-LL
(Fig.7), both models on average tracked the instrumental record quite well with
statistically higher correlation coefficients with observation (HadCRUT5). That is,
improvements in $SO_2$ deposition parameterizations have contributed to better model
performances in reproducing historical surface temperature evolution.

**3.5 Relative changes preceding and following the 1960-1990 period**

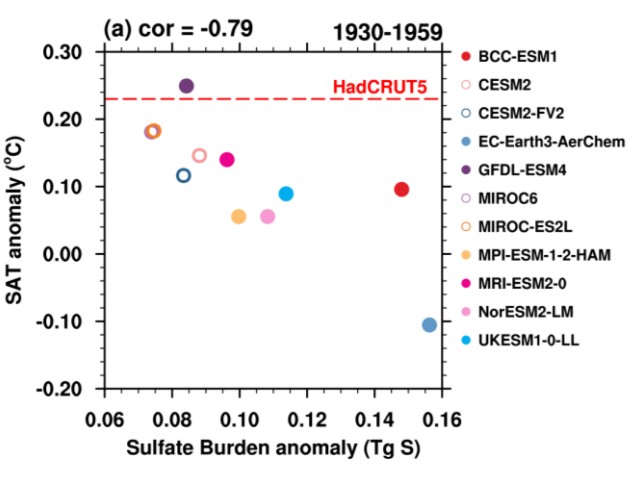

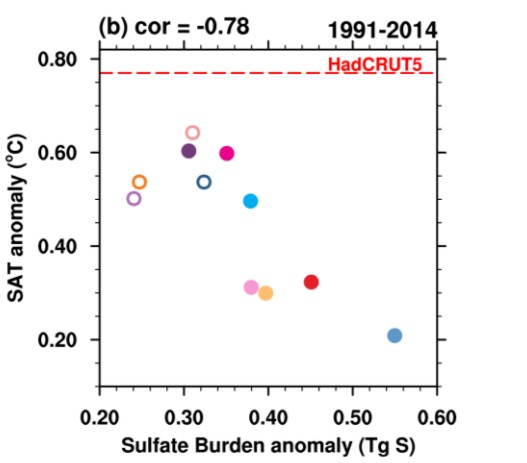


**Figure 8.** Scatter plots of sulfate burden anomalies versus SATa in (a) 1930-1959, and (b) 1991-

348    2014.

Our analysis reveals a robust correlation between sulfate burden anomalies and
SATa during 1960-1990 (Fig. 2a). To evaluate the temporal consistency of this

relationship, we examined its behavior before and after this period. Given that the relationship reflects clear underlying physics, similar correlations were expected across different periods. As shown in Fig.8, statistically significant correlations are evident in both periods, suggesting that sulfate burden anomalies were overestimated prior to 1960-1990, and this overestimation continued to influence SATa in subsequent decades. Compared to HadCRUT5, the models on average underestimate SATa by 0.11°C during 1930-1959 and by 0.31°C during 1991-2014. The correlations between sulfate burden anomalies and SATa are -0.79 and -0.78 for these two periods, respectively, which are weaker than the correlation of -0.91 during 1960-1990. This weakening may be partly attributable to the smaller biases in the 1930-1959 interval. Furthermore, the combined effects of increasing atmospheric $CO_2$ concentrations since the Industrial Revolution and the high climate sensitivity in CMIP6 models may have partially offset the cooling bias during 1991-2014 (Hausfather et al., 2022).

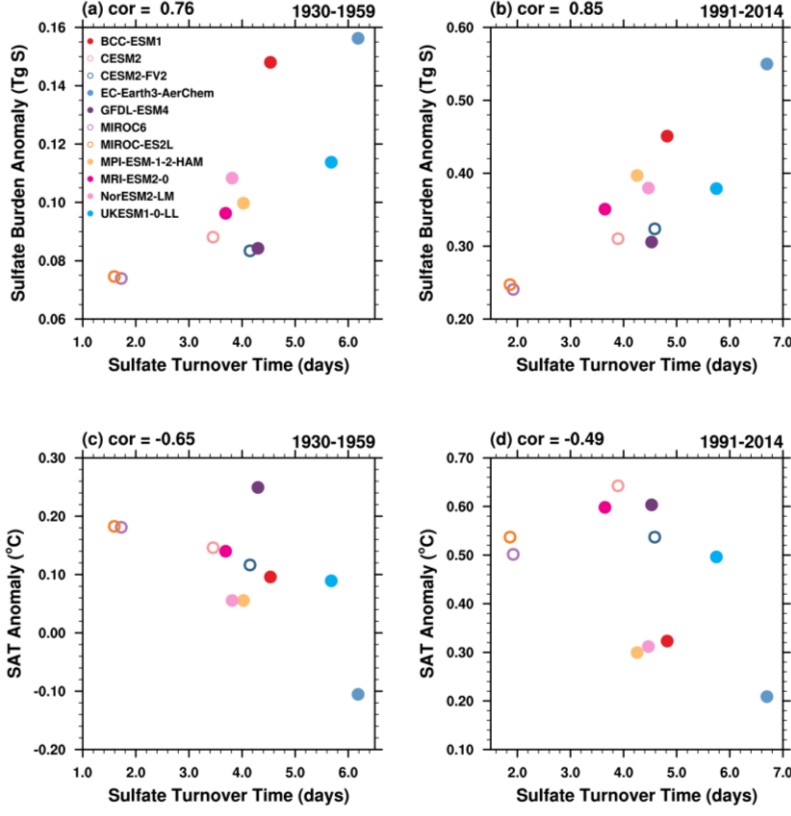

**Figure 9.** Sulfate turnover time ($\tau_{SO_4}$) versus (a, b) sulfate burden anomalies, and (c, d) SATa for the periods 1930-1959 and 1991-2014.

Sulfate turnover time is a key parameter governing sulfate burden and shows
strong correlations with sulfate burden anomalies and SATa during 1960-1990 (Figs.
6b and 6c). Statistically significant correlations persist before and after this period (Fig.
9), confirming the dominant role of sulfate-related physical processes across all
examined time intervals.

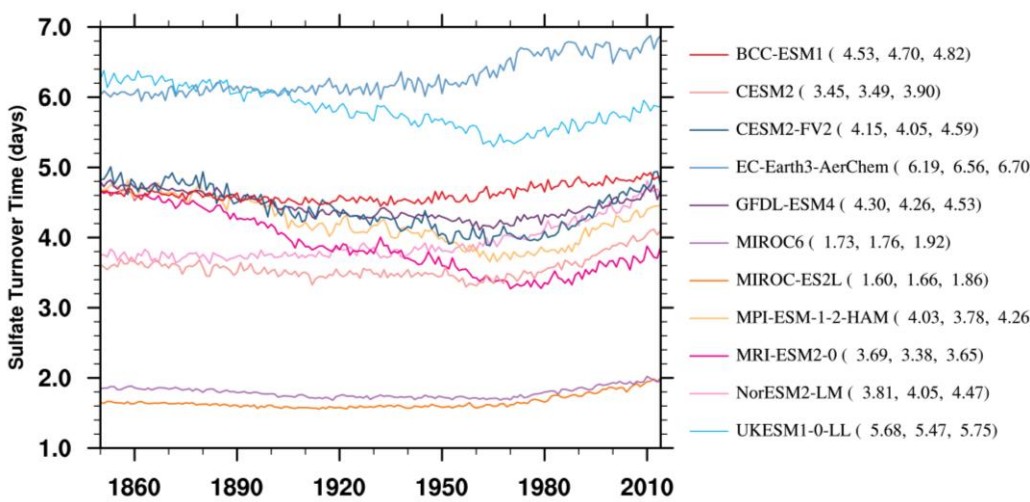


**Figure 10.** Temporal evolution of sulfate turnover time ($\tau_{SO_4}$) in CMIP6 models. Numerical labels
denote mean $\tau_{SO_4}$ value during 1930-1959, 1960-1990, and 1991-2014.

We also analyze the temporal evolution of sulfate turnover time (Fig.10). Its
temporal variability, characterized by a standard deviation ($\sigma < 0.5$ days), is notably
smaller than the inter-model spread. During 1930-1959, models exhibit a divergent
trend, with 5 out of 11 models simulating reduced turnover times in the subsequent
period. In contrast, all models show prolonged turnover times during 1991-2014
compared to earlier periods. This shift may be partly attributable to changes in the
regional distribution of sulfur emissions, including an increasing proportion of
emissions from Asia and the implementation of stringent emission control policies in
Europe and North America.

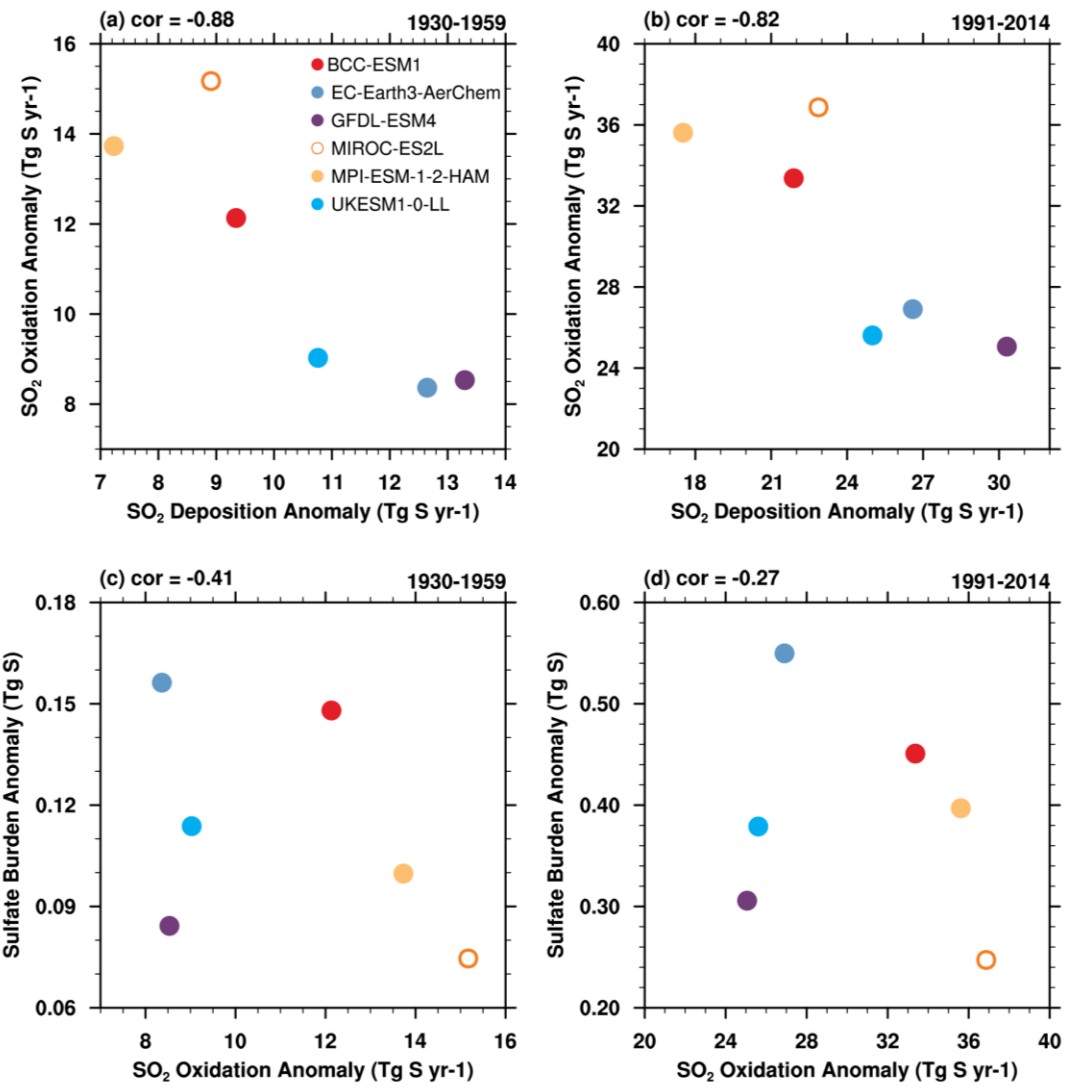

**Figure 11.** Same as Fig.4, but for (a, c) 1930-1959, and (b, d) 1991-2014.

SO₂ deposition maintains a strong negative correlation with SO₂ oxidation both before and after the 1960-1990 period (Fig.11), with coefficients of -0.88 and -0.82, respectively. Meanwhile, the anomaly in SO₂ oxidation exhibits a negative but statistically insignificant correlation with the sulfate burden anomaly.

## 4. Conclusions

The aerosol cooling effect is considered as the second most important anthropogenic forcing during the 20[th] Century. Based on the 11 CMIP6 models with interactive aerosol schemes, our study demonstrates that the cooling bias during 1960-

1990 is closely related to the sulfate burden changes in the atmosphere. Sulfate aerosol represents the terminal product of a complex chain of physicochemical processes that convert sulfur emissions into sulfate particles. Our findings indicate that sulfate burden anomalies in these models are governed by two key processes: the removal of its gaseous precursor $SO_2$ and sulfate deposition itself. Higher $SO_2$ deposition rates limit the availability of $SO_2$ for subsequent oxidations. Sulfate turnover time is critical for evaluating the physical realism of models. Comparative analysis with observational measurements reveals that increasing sulfate deposition to reduce sulfate burden anomalies is not a reasonable approach. Biases in sulfate burden anomalies may be driven by discrepancies in simulating upstream $SO_2$ deposition and oxidation processes, rather than downstream processes. This is further supported by improvements in two post-CMIP6 models with refined $SO_2$ deposition parameterizations.

Analyses for periods preceding and following 1960-1990 confirm the persistent influence of sulfate-related physical processes across all examined time periods. Therefore, CMIP6 model projections should be interpreted with caution, as they may underestimate future warming rates. It is therefore also essential to evaluate the reliability of sulfate-related processes in upcoming model intercomparisons before applying them to future climate projections. We encourage future intercomparison initiatives to archive sulfur cycle relevant outputs from a wider range of participating models, thereby enabling more robust and comprehensive process-oriented evaluations.

**Code availability**

All data processing codes are available if a request is sent to the corresponding authors.

**Data availability**

The HadCRUT5 dataset is accessible through the Met Office Hadley Centre observations database (https://www.metoffice.gov.uk/hadobs/hadcrut5/). All the model data can be freely downloaded from the Earth System Grid Federation (ESGF) nodes (https://aims2.llnl.gov/search/cmip6/).

425

**Author contributions**

The main ideas were formulated by J.Z. and K.F. J.Z. wrote the original draft. The results were supervised by K.F. and S.T.T. All the authors discussed the results and contributed to the final manuscript.

430

**Competing interests**

The authors declare no competing financial and/or non-financial interests.

433

**Acknowledgements**

We would like to thank the editors for handling the manuscript and providing constructive guidance throughout the review process. We sincerely appreciate the insightful comments from Dr. Stephen E. Schwartz and the anonymous reviewers, which significantly improved the quality of this work. We acknowledge all data developers, their managers, and funding agencies for the datasets used in this study, whose contributions and support were essential to our research.

441

**Financial support**

This work was jointly supported by the National Natural Science Foundation of China (Grant no. U2542212 and 42230608) and the UK–China Research and Innovation Partnership Fund through the Met Office Climate Science for Service Partnership (CSSP) China as part of the Newton Fund.

447

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
