# Peer review of "Unveiling the Dominant Control of the Systematic Cooling Bias in"

_EGUsphere, 2025_

## Referee Comment (RC2)

Review of: Unveiling Sulfate Aerosol Persistence as the Dominant Control of the Systematic Cooling Bias in CMIP6 Models: Quantification and Corrective Strategies by Jie Zhang et al, for ACP.

Stephen E. Schwartz, Reviewer; 2025-0513

The authors make the point that "Including sophisticated aerosol schemes in the models of the sixth Coupled Model Inter-comparison Project (CMIP6) has not improved historical climate simulations." They go on to state that "in particular, the models underestimate the surface air temperature anomaly (SATa) when anthropogenic sulfur emissions increased in 1960~1990, making the reliability of the CMIP6 projections questionable."

The present paper seems to consist of two components: (1) The differences among models in the sulfate forcing and (2) the relation of sulfate forcing to temperature change among the models.

Before going further in this review, I must state major disagreement with the authors' definition of what they denote as the "effective retention time" of atmospheric sulfur. This quantity is found in the authors' analysis of the CMIP model results to be as low as 1 day or so. In my judgment this flawed definition casts a cloud over all the findings in the study. This is elaborated below.

In the present study the authors define and use a quantity that they call the ESRT, effective sulfur retention time scale, equation 1, line 295.

$$ESRT = loadSO4/(D_{SO_4}+D_{SO_2}),$$

This quantity is central to the author's analysis so in my opinion should not be hidden back in the appendix and justified. I disagree with the utility of this definition. It is a stock of sulfate upon leaving flux of atmospheric sulfate plus leaving flux of atmospheric SO2. This is contrary to customary definition of residence time (more explicitly, turnover time, to distinguish it from other measures of residence time) which would be stock of the substance of interest in a given compartment upon the leaving flux of that substance from that compartment, not upon multiple fluxes of multiple substances; the definition confounds the utility of the definition and may be

responsible for the low residence time found in the study. The definition certainly would seem to preclude the utility of any comparisons with residence time as conventionally evaluated.

Setting aside that concern, Figures1b, 4a and 4b seem to show substantial systematic differences across models in measures of sulfate burden that might provide insight into reasons for differences in sulfate forcing across models.

With respect to the relation, across models, between sulfate burden and change in GMST, the authors seem unaware of the extensive prior literature on this subject that is very pertinent to this study. In early work Kiehl (2007) demonstrated anticorrelation between forcing and sensitivity in the climate model runs reported in the 2007 IPCC report. The relation between sulfate forcing and temperature change in models continues through Stevens (2015) and beyond. Certainly, it would seem that any revision of the current paper should take cognizance of this literature.

[Figure]

From Kiehl (2007)

The role of sulfate lifetime as a key influence on forcing, which is a major thesis of this paper, was highlighted by Charlson et al, 1992, who delineated the factors controlling direct sulfate forcing, as given in the following table:

**Table 1.** Evaluation of global-mean direct radiative forcing due to anthropogenic sulfate (Eqs. 1 through 5). Double underbar indicates evaluation from preceding quantities.

| Quantity | Value | Units | Relative uncertainty (%) | Reference |
|---|---|---|---|---|
| $Q_{SO_2}$ | $90 \times 10^{12}$ | g of sulfur per year | 15 | * |
|  | $2.8 \times 10^{12}$ | mol of sulfur per year |  |  |
| $Y_{SO_4^{2-}}$ | 0.4 |  | 50 | † |
| $\tau_{SO_4^{2-}}$ | 0.02 | year | 50 | ‡ |
| $A$ | $5 \times 10^{14}$ | m² |  |  |
| $B_{SO_4^{2-}}$ | $4.6 \times 10^{-3}$ | $(g\ SO_4^{2-})\ m^{-2}$ |  |  |
|  | $4.8 \times 10^{-5}$ | mol m$^{-2}$ |  |  |
| $\alpha_{SO_4^{2-}}$ | 5 | $m^2\ (g\ SO_4^{2-})^{-1}$ | 40 | (22, 37, |
|  | $4.8 \times 10^2$ | $m^2\ mol^{-1}$ |  | 45, 58) |
| $f(RH)$ | 1.7 |  | 20 | (44, 45) |
| $\delta_{SO_4^{2-}}$ | 0.04 |  |  |  |
| $F_T$ | 1370 | W m$^{-2}$ |  |  |
| $T$ | 0.76 |  | 20 |  |
| $1 - A_c$ | 0.4 |  | 10 | (59) |
| $1 - R_s$ | 0.85 |  | 10 | (60) |
| $\overline{\beta}$ | 0.29 |  | 25 | (61) |
| $\overline{\Delta F_R}$ | −1.3 | W m$^{-2}$ |  |  |

*From Fig. 1.   †Mean dry-deposition velocity of $SO_2$, 0.5 cm s$^{-1}$; height of mixed layer, 2 km; $k_{SO_2\text{-OH}}$, $1 \times 10^{-12}$ cm³ s$^{-1}$; mean OH concentration, $1 \times 10^6$ cm$^{-3}$; in-cloud conversion assumed equal to that of gas phase (45). These parameter values are consistent with results of more detailed calculations [for example, (57)].   ‡$\tau_{SO_4^{2-}}$, 1 week, representing the frequency of encountering precipitation removal.

Those investigators also presented an analysis of indirect (Twomey) forcing that rested on mass loading of atmospheric sulfate, which in turn is proportional to sulfate residence time. Unless the authors take exception to this sort of analysis, I would suggest that this might be a good place to start in their assessment; if they do take exception, then it would seem incumbent on them to identify the point of disagreement and to justify their concern. The quantity $\tau_{SO_{42-}}$ in the above is

0.02 yr or 7.3 days. This value of $\tau_{SO4^{2-}}$ was based in in that study in large part on the observed rate of decrease of concentration of 137Cs at several midlatitude stations in the Northern Hemisphere subsequent to the release of this isotope from the Chernobyl incident in 1986. The negative of the slope of the log of that concentration vs time gives the decay constant; the inverse of that, the residence time. 137Cs released by the reactor release attaches to accumulation mode particles and thus serves as a good proxy for determination of the lifetime of accumulation-mode particles or sulfate.

[Figure]

Modified from Cambray et al. (1987)

Subsequently, the incident at the Fukushima reactor following the 2011 tsunami provides even better information on lifetime from the ratio of 137Cs to the passive tracer 133Xe (Kristiansen et al, 2012) which removes much of the variability in concentration of a single species arising from variability in transport. That analysis yields a residence time of about 13 days.

[Figure]

All individual stations
$y = 1.8e{-}005 \exp(-t/13.9)$
$R^2 = 0.38*$

Daily averages all stations ▽
$y = 4.0e{-}005 \exp(-t/12.6)$
$R^2 = 0.65*$

| | |
|---|---|
| ● | wakeisland |
| ● | oahu |
| ● | ulanbator |
| ● | ussuriysk |
| ● | ashland |
| ● | charlottesville |
| ● | yellowknife |
| ● | stjohns |
| ● | schauinsland |
| ● | stockholm |
| ● | spitsbergen |

$^{137}Cs / ^{133}Xe$

[days]

Kristiansen et al ACP 2012

In sum, the observational evidence would seem to contradict the short residence time presented by the authors based on their analysis of the CMIP models

Several prior model intercomparisons have been conducted that identify the loading of sulfate (and other aerosol constituents) to try to identify reasons for differences in forcings. I call particular attention to the AeroCom Study (Kinne et al, 2006; Textor et al., 2006). Textor examined residence times of multiple aerosol constituents in a very insightful figure. The model calculations were found to give a much greater residence time for sulfate (3-5.5 days) than reported for the CMIP models as reported in the present manuscript. Stevens (2015) revisited the quantities employed by Charlson et al in their 1992 study, specifically including sulfate lifetime, for which he accepted the results from the model studies taking the value of 3.8 days.

[Figure]

Tropospheric residence times in [days] in the AeroCom models for the species under consideration. Textor et al., 2006.

Whether the short lifetime presented by the authors in the present analysis (about 1 day) is an accurate representation of the lifetime of sulfate in the CMIP models or is erroneous due to their definition cannot be discerned from the manuscript. But it would seem that without further justification of Eq 1, that definition would seem to be fatal to the analysis presented.

Given the concerns raised above I conclude my review here.

**References**

Cambray, R.S., Cawse, P.A., Garland, J.A., Gibson, J.A.B., Johnson, P., Lewis, G.N.J., Newton, D., Salmon, L. and Wade, B.O., 1987. Observations on radioactivity from the Chernobyl accident. Nucl. Energy, 26(2), pp.77-101.

Charlson, R.J., Schwartz, S.E., Hales, J.M., Cess, R.D., Coakley Jr, J.A., Hansen, J.E. and Hofmann, D.J., 1992. Climate forcing by anthropogenic aerosols. Science, 255(5043), pp.423-430.

Kiehl, J.T., 2007. Twentieth century climate model response and climate sensitivity. Geophysical Research Letters, 34(22).

Kinne, S., Schulz, M., Textor, C., Guibert, S., Balkanski, Y., Bauer, S.E., Berntsen, T., Berglen, T.F., Boucher, O., Chin, M. and Collins, W., 2006. An AeroCom initial assessment–optical properties in aerosol component modules of global models. Atmospheric Chemistry and Physics, 6(7), pp.1815-1834.

Kristiansen, N.I., Stohl, A. and Wotawa, G., 2012. Atmospheric removal times of the aerosol-bound radionuclides 137 Cs and 131 I measured after the Fukushima Dai-ichi nuclear accident–a constraint for air quality and climate models. Atmospheric Chemistry and Physics, 12(22), pp.10759-10769.

Stevens, B., 2015. Rethinking the lower bound on aerosol radiative forcing. Journal of Climate, 28(12), pp.4794-4819.

Textor, C., Schulz, M., Guibert, S., Kinne, S., Balkanski, Y., Bauer, S., Berntsen, T., Berglen, T., Boucher, O., Chin, M. and Dentener, F., 2006. Analysis and quantification of the diversities of aerosol life cycles within AeroCom. Atmospheric Chemistry and Physics, 6(7), pp.1777-1813.

---

## Author Comment (AC2)

Replies to Dr. Stephen E. Schwartz's review on: Unveiling Sulfate Aerosol Persistence as the Dominant Control of the Systematic Cooling Bias in CMIP6 Models: Quantification and Corrective Strategies by Jie Zhang et al, for ACP.

Jie ZHANG, 2025-0605

Thank Dr. Schwartz for your constructive critique, particularly for emphasizing the importance of distinguishing the "effective sulfur residence time" (ESRT) from other lifetime measures reported in the literature. We acknowledge that the term "effective sulfur residence time (ESRT)" is potentially misleading.

ESRT was envisioned primarily as a diagnostic tool (not a physical timescale) for model tuning. Because it accounts for both sulfate and $SO_2$ deposition, its value is typically lower than the sulfate atmospheric lifetime. As Dr. Schwartz noted, it is fundamentally a metric for model evaluation rather than a conventional definition of atmospheric residence time and should not be interpreted as such. Therefore, we propose renaming it to the "Sulfur Assessment Metric for ESMs" (SAME).

We acknowledge that sulfate lifetime remains critical for validating the model's physical realism. Therefore, we calculated sulfate lifetime as the ratio of sulfate burden to total sulfate deposition (wet plus dry) in the CMIP6 models, BCC-ESM1-1, and UKESM1-1-LL (Table A1). Sulfate lifetime ranges from 1.65 days in MIROC models to 6.57 days in EC-Earth3-AerChem, which is consistent with previous literatures, particularly the estimates in AeroCom models (Textor et al., 2006). This wide range is attributed to variations in both sulfate burden (0.33 to 0.75 Tg S) and deposition rates (0.75 to 7.58 Tg S $yr^{-1}$ for dry deposition and 31.68 to 69.41 Tg S $yr^{-1}$ for wet deposition).

**Table 1** Sulfate burden, sulfate depositions, and sulfate lifetime in CMIP6 models, BCC-ESM1-1 and UKESM1-1-LL in PHC period.

| model name | Sulfate burden (Tg S) | Sulfate Deposition (Tg S yr-1) | | Sulfate lifetime (days) |
|---|---|---|---|---|
| | | DrySO4 | WetSO4 | |
| BCC-ESM1 | 0.59 | 2.07 | 43.78 | 4.70 |
| CESM2 | 0.40 | 6.14 | 35.13 | 3.54 |
| CESM-FV2 | 0.43 | 5.92 | 32.60 | 4.07 |
| EC-Earth3-AerChem | 0.75 | 1.29 | 40.39 | 6.57 |
| GFDL-ESM4 | 0.46 | 7.58 | 31.68 | 4.28 |
| MIROC6 | 0.33 | 7.50 | 61.29 | 1.75 |
| MIROC-ES2L | 0.33 | 5.67 | 67.42 | 1.65 |
| MPI-ESM-1-2-HAM | 0.74 | 2.41 | 69.41 | 3.76 |
| MRI-ESM2-0 | 0.53 | 0.75 | 56.96 | 3.35 |
| NorESM2-LM | 0.52 | 6.39 | 40.33 | 4.06 |
| UKESM1-0-LL | 0.63 | 7.00 | 34.79 | 5.50 |
| BCC-ESM1-1 | 0.48 | 1.34 | 19.2 | 8.53 |
| UKESM1-1-LL | 0.52 | 5.57 | 27.34 | 5.77 |

Sulfate lifetimes in BCC-ESM1-1 (8.53 days) and UKESM1-1-LL (5.77 days) are generally longer than in their previous versions (4.70 and 5.50 days, respectively). The longer sulfate lifetimes in the updated models may be due to lower $SO_2$ in these revised models but also could be due to physical climate changes (e.g., temperatures, clouds, rainfall). Compared to prior lifetime measures reported in the literature and considering the range of lifetimes found in recent models, the sulfate lifetimes in BCC-ESM1-1 and UKESM1-1-LL appear reasonable (e.g., Charlson et al, 1992; Kristiansen et al. 2012; Textor et al., 2006).

[Figure]

**Figure A1.** (a) Scatter plots of yearly total sulfur sink anomaly (x-axis, Tg S yr-1) versus sulfate burden anomaly (y-axis, Tg S) in PHC period in relative to 1850~1900 mean. Number in legend shows the mean and standard deviation of ratio between sulfate burden anomaly and total sulfur sink anomaly in PHC period, defined as SAME, units: days. (b) The SATa (°C, x-axis) versus SAME (days, y-axis) in PHC period for each model. The black solid line is the linear fitting. The blue and red solid lines are the 95% confidence interval (CI) and 95% prediction interval (PI), respectively. SAT anomaly in HadCRUT5 and its 0.175°C boundaries are shown by the red dashed line and parallel gray dashed lines, respectively. The red and blue asterisks are the results in the two post-CMIP6 models BCC-ESM1-1 and UKESM1-1-LL, respectively.

To eliminate the effect of differing climatological states across models, the SAME metric is defined as the ratio of sulfate anomaly during the PHC period to the sum of sulfate and $SO_2$ deposition anomalies:

SAME = loadSO4a / (Daso4 + Daso2)

where:

- loadSO4a is the total sulfate loading anomaly in the atmosphere,

- Daso4 denotes the total (wet plus dry) sulfate deposition anomaly, and

- Daso2 denotes the total (wet plus dry) $SO_2$ deposition anomaly during the PHC period.

The SAME ranges from 1.1 days in MIROC models to 2.86 days in EC-Earth3-AerChem (Fig. A1a). The correlation coefficient between SATa and SAME is -0.90 (Fig. A1b). In addition to the linear regression between SATa and SAME (black line in Fig. A1b), we also show the 95% confidence interval (CI, blue curves) and the 95% prediction interval (PI, red curves).

There are 4 models with SAT values around 0.165°C (the lower limit of observation), giving a range of SAME between 1.1 to 1.58 days. Since most models underestimate SATa, it is difficult to predict SAME values when SATa is higher than the observation (0.34°C). Results from BCC-ESM1-1 suggest that SAME may not need to decrease at the regression line rate beyond the lower limit (0.165 °C). Therefore, we recommend a SAME value of 1.35 ± 0.25 days based on the 95% CI (encompassing the four CMIP6 models and BCC-ESM1-1), or a wider range of 1.35 ± 0.6 days by the 95% PI that includes the UKESM1-1-LL.

Generally, the SAME metric is used to facilitate model tuning of the aerosol load, ensuring that models do not overestimating the aerosol cooling effect over the historical period, as was the case in CMIP6 and is a current concern for model performance in the upcoming CMIP7 experiments. To ensure model credibility, the sulfate atmospheric lifetime must align with observations (literature) values, which

our calculations above confirm. The manuscript will be amended to include both the sulfate lifetime in CMIP6 models and the two updated models, along with the SAME metric and its recommendations for model performance.

---

## Author Response (AR1)

In the following, the text with italicization indicates the Reviewers' comments, and the normal text is our response.

**Replies to Reviewer's comments:**

*Reviewer(s)' Comments to Author(s):*
*Reviewer: 1*

*The manuscript firstly points out that most CMIP6 earth system models (with interactive atmospheric sulfate cycle) present cold biases in the period 1960-1990 (called PHC in the manuscript, pot-hole cooling). The authors performed then a series of investigations searching the relation of this cold bias with sulfate sources and sinks across the available CMIP6 models. The authors finally proposed a single parameter "ESRT", effective sulfate retention time. This is an interesting diagnostic, relatively stable for a given model and quite useful to characterize its sulfate cycle. It was shown that ESRT has a good capacity to explain the cold bias across models. It is also interesting to see that the authors use the temperature anomalies of the PHC period to "constrain" the optimal value of ESRT. This optimal value is then used to approximate the "right" sulfate deposition rate which is furthermore used in the BCC model with improved performance.*

*All that said, I have a small concern for what shown in Fig. 1a displaying temperature time series. From those curves, I can deduce that the cold bias of models in the PHC period is not exceptional, not as the authors pointed out, since there is a good trend compared to observation. But the cold bias (at least in the multi-model ensemble mean) occurred before the PHC period, roughly at the point of 1935 where models drift significantly from observation and the cold bias remains for the rest of the time, including the PHC period (1960-1990).*

[Figure]

**Figure 12.** Ensemble mean historical surface warming in CMIP5 and CMIP6 compared with observations. Shading on the models is the ensemble SD. The baseline is 1850–1900.

https://doi.org/10.5194/acp-20-7829-2020

**Response:** Thank you for your comments. We reference Figure 12 from Flynn and Mauritsen (2020), which evaluates historical surface temperature anomalies in CMIP5 and CMIP6 models. Their analysis indicates that the CMIP5 multi-model ensemble mean effectively captured the instrumental record with observation falling well within model spread – a consistency also noted in CMIP3 models assessed in the IPCC Third Assessment Report (IPCC AR3). In contrast, a majority of CMIP6 models exhibit a cold bias in surface temperature, marking a notable departure from earlier model generations. We clarify this in L146-149: "The anomalous cooling in CMIP6 model marked a notable departure from earlier model generations, which can effectively capture the instrumental SAT record with observation falling well within model spread (e.g., Flynn and Mauritsen, 2020; Hegerl, et al., 2007)."

You are right. The cold bias occurred before the PHC period, roughly at the point of 1935. We think it can also be attributed to elevated sulfate aerosol burdens. As shown in Fig.1b, the sulfate burden increased steadily since the Industrial Revolution. The selection of the 1960–1990 as PHC period in our analysis stems from its alignment with accelerated anthropogenic emissions, particularly of sulfate precursors (e.g., $SO_2$). The large anthropogenic emissions in PHC amplify the model-observation divergence during this era. By focusing on this interval, we aim to quantify the climate impacts of anthropogenic aerosols during a period of rapidly increasing industrial activity. We clarify this in section 3.1 (L145-L146): "All the models tend to underestimate SATa since the 1930s.", L150: "The cooling bias is most pronounced from 1960 to 1990, i.e., the PHC period.", and in L169-172: "The PHC coincides with increased anthropogenic emissions, particularly of sulfate precursors such as $SO_2$ (Zhang et al., 2021a). Global emissions of $SO_2$ grew steadily after the 1950s and peaked in the 1970s at 180Tg yr$^{-1}$, which is about 3.6 times the 1950s' emissions (Hoesly et al., 2018)."

**Replies to Dr. Stephen E. Schwartz's review on:** Unveiling Sulfate Aerosol Persistence as the Dominant Control of the Systematic Cooling Bias in CMIP6 Models: Quantification and Corrective Strategies by Jie Zhang et al, for ACP.

In the following, the text with italicization indicates Dr. Stephen E. Schwartz's comments, and the normal text is our response.

*Comments to Author(s):*

*I have major concern over the definition of the quantity that the authors call the ESRT, effective sulfur retention time scale. This is a non-conventional definition of a residence time that may account for the short (ca 1 day) values reported, and certainly precluding comparison with other measures of lifetime in the literature.*

*The authors seem unaware of the large prior literature pertinent to this study.*

*I elaborate on these concerns in the pdf review.*

**Response:**

Thank Dr. Schwartz for your constructive critique, particularly for emphasizing the importance of distinguishing the "effective sulfur residence time" (ESRT) from other lifetime measures reported in the literature. We acknowledge that the term "effective sulfur residence time (ESRT)" is potentially misleading.

ESRT was envisioned primarily as a diagnostic tool (not a physical timescale) for model tuning. Because it accounts for both sulfate and $SO_2$ deposition, its value is typically lower than the sulfate atmospheric lifetime. As noted, it is fundamentally a metric for model evaluation rather than a conventional definition of atmospheric residence time. Therefore, we renaming it to the "Sulfur Assessment Metric for ESMs" (SAME) in the revised manuscript.

We acknowledge that sulfate lifetime remains critical for validating the model's physical realism. And there are three major changes in the revised manuscript according to Dr. Schwartz's comments. The manuscript is amended accordingly based on these three major modifications, including the introduction.

**1. Discussion about sulfate lifetime in Section 4.**

We calculated sulfate lifetime as the ratio of sulfate burden to total sulfate deposition (wet plus dry) in the CMIP6 models, BCC-ESM1-1, and UKESM1-1-LL (Table 2) to ensure model credibility. The detailed analysis and discussion are presented in the newly added Section 4: "**Discussion: Sulfate lifetime in CMIP6 models and the two post-CMIP6 models**".

The analysis and discussion are shown in L348-L363: "As shown in Table 2, sulfate lifetime in CMIP6 models ranges from 1.65 days in MIROC-ES2L to 6.57 days in EC-Earth3-AerChem. The mean sulfate lifetime is 3.93 days, consistent with previous literatures, particularly the mean value of 4.12 days in AeroCom models with standard deviation of 18% (Textor et al., 2006). The wide sulfate lifetime range in CMIP6 models is attributed to variations in both sulfate burden (0.33 to 0.75 Tg S) and deposition rates (0.75 to 7.58 Tg S $yr^{-1}$ for dry deposition, and 31.68 to 69.41 Tg S $yr^{-1}$ for wet deposition).

Sulfate lifetimes in the two post-CMIP6 models, 8.53 days in BCC-ESM1-1 and 5.77 days in UKESM1-1-LL, are generally longer than those of their CMIP6 versions. The longer sulfate lifetimes in the two post-CMIP6 models may be due to lower SO2 in these revised models but also could be due to physical climate changes (e.g., temperatures, clouds, rainfall). Compared to prior lifetime measures reported in the literature and considering the range of lifetimes found in recent models, the sulfate lifetimes in BCC-ESM1-1 and UKESM1-1-LL also appear reasonable (e.g., Charlson et al, 1992; Kristiansen et al. 2012; Textor et al., 2006)."

**2. Definition of SAME index.**

To eliminate the effect of differing climatological states across models, in the revised manuscript the SAME metric is defined as the ratio of sulfate anomaly during the PHC period to the sum of sulfate and $SO_2$ deposition anomalies:

SAME = loadSO4a / (DSO4a + DSO2a)

where:

- loadSO4a is the total sulfate loading anomaly in the atmosphere,

- DSO4a denotes the total (wet plus dry) sulfate deposition anomaly, and

- DSO2a denotes the total (wet plus dry) $SO_2$ deposition anomaly during the
PHC period.

Since the definition of SAME is central to our analysis, we add a new section
(**Section 2: Model, data, and method**) to introduce the data and methods as suggested.

**3. Refine the constraint of SAME.**

As shown in Fig.4a, the SAME ranges from 1.1 days in MIROC models to 2.86
days in EC-Earth3-AerChem. The correlation coefficient between SATa and SAME is
-0.90 (Fig. 4b). In addition to the linear regression between SATa and SAME (black
line in Fig. 4b), **in the revised manuscript, we also calculate the 95% confidence
interval (CI, blue curves) and the 95% prediction interval (PI, red curves):**" (L292-
294) We calculate the linear fitting between SATa and SAME (black line in Fig. 4b),
the 95% confidence interval (CI, blue curves), and the 95% prediction interval (PI, red
curves), respectively.",   "(L303-310)As shown by the red asterisk in Fig.4b, the
SAME reduced from 2.51 days to 1.43 days in updated BCC-ESM1 (BCC-ESM1-1),
falling right within the PI constraint. The new SAME index is 57% of its previous
values. Accordingly, the SATa in PHC is 0.34°C, falling within the observational range
from 0.165°C to 0.515°C. We also examine the SAME in UKESM1-1-LL with modified
$SO_2$ dry deposition parameterization. The SAME is shortened from 2.19 days to 1.71
days, falling within the CI constraint. Accordingly, the SATa in PHC period increases
by about 0.25°C."

We also discuss the uncertainty in SAME estimate in L311-322: "Given that most
models underestimate SATa relative to observations, extrapolating SAME values for

SATa exceeding the observation ($0.34^{\circ}$C) becomes highly uncertain. Result from BCC-ESM1-1 suggests that the rate of decrease in SAME predicted by the regression line may not hold for SATa values above the observed lower bound ($0.165$ °C). Therefore, we recommend a central SAME estimate of 1.35 days. Critically, this value carries inherent uncertainties that must be quantified:

- The 95% confidence interval (CI) of ±0.25 days (i.e., 1.10–1.60 days).

- The wider 95% prediction interval (PI) of ±0.6 days (i.e., 0.75–1.95 days).

The substantial difference between the CI and PI ranges underscores the challenge in precisely constraining SAME. We advise using the PI for applications requiring robustness against individual model deviations."

---

## Referee Report (RR1)

Review of: Unveiling Sulfate Aerosol Persistence as the Dominant Control of the Systematic Cooling Bias in CMIP6 Models: Quantification and Corrective Strategies by Jie Zhang et al, for ACP. Version 2

Stephen E. Schwartz, Reviewer; 2025-0715; modified 2025-0716

I remain unpersuaded by the changes in this manuscript relative to Version 1.

Rather than change the definition of the quantity previously denoted Effective Sulfate Retention Time ESRT to something like a turnover time for a species, or yield for sulfate, which would have physical meaning, the authors simply changed the name of this quantity to Sulfur Assessment Metric for Earth System model evaluation SAME. Changing the name of a physically non-meaningful quantity does not make it physically meaningful.

I think that the key extensive measure of sulfate influence would simply be the sulfate burden, as that is the quantity that does the forcing, or the sulfate forcing itself. Burden anomaly vs SAT anomaly is shown in Fig 2a (Better the other way around, Burden anomaly being the independent variable and SAT anomaly being the dependent quantity). This burden is a consequence of model parameters that govern formation and removal, such as reaction coefficients and removal coefficients, or the SO2 emission rate.

Given the anticorrelation between change in SAT and sulfate burden, the authors are evidently trying to find the reason for the differences in sulfate burden across the models. This is laudable. Such difference might be due to different emissions, different sulfate yield per emitted SO2 or different sulfate lifetime (turnover time) or some combination. The latter two are physically meaningful intensive properties of the system. I argued previously that the ESRT, now renamed SAME, although an intensive property, is not a physically meaningful quantity. Figure 4a seems to make a strong case that the differences in sulfate burden among models is due to different total SO2 + sulfate sink anomaly, the slope being ESRT, now renamed SAME, which differs for the different models. But showing this correlation does not point to a path forward in terms of assessing the accuracy of representation of processes in models. Modelers cannot directly modify representation of SAME in their models because SAME is not a physical quantity.

It would also seem essential to compare the relative values of SO2 sink rate and sulfate sink rate; does it make sense to add these quantities if one or the other is dominant? Are the differences across models due to differences in sulfate sink rate or SO2 sink rate? These questions speak to differences in the consequences of different representations of processes. ESRT, now renamed SAME, does not.

One wonders what a plot of sulfate burden anomaly vs sulfate sink rate anomaly like Fig 4a would look like; this slope would be sulfate turnover time, a physically meaningful quantity that can be compared to measurements such as the studies by Cambray and by Kristiansen that I called attention to in my previous review.

One wonders whether there is appreciable difference between such plots during PHC and non PHC periods. The quantities are intensive, depending on the same representation of physical processes, so in principle they depend on the governing physics, not on whether the data are from the PHC or non-PHC time periods. If the slopes are different in non PHC periods, that would be interesting. That would direct attention to representation of processes in models.

The authors should be able to evaluate sulfate yield (rate of production of sulfate/rate of SO2 emission) across the models. Again, this is a physically meaningful quantity. Is there appreciable difference across the models.

If there is little correlation across the models between sulfate burden vs yield, and between sulfate burden vs sulfate dep rate, that would suggest some anticorrelation across the models between yield and sulfate lifetime, which would be an important finding.

The authors state (abstract) that that their metric SAME is overestimated by almost all the CMIP6 models. One would ask what is the basis for this statement. Overestimated relative to what standard?

Minor comment: I wonder whether the terms "anomalous sulfate deposition rate" etc. might be replaced by "sulfate deposition rate anomaly" etc. throughout the text, as used in figure axis labels, and consistent with commonly used "temperature anomaly"

Minor comment: I do not care for the terminology "pot hole cooling period" (PHC). A simple reference to the time period 1960-1990, during which GMST decreased (or perhaps better, did not exhibit same rate of increase as in earlier and later times). Language matters.

Minor comment. Equations, such as the one given for SAME should be written with algebraic symbols, not words.

Important comment for authors and editor, should this paper reach the acceptance stage: Tables should be presented of all graphical quantities such as time series, as in Fig 1, and x-y plots as in Figs 2-4, in an appendix or supplementary material. These are highly processed data. It is wholly unacceptable simply to state that the data can be freely downloaded from the EGSF nodes.

*Some further thoughts on the manuscript 2025-0716*

In my review I did not pay much attention to the assumption inherent in Fig 2 that the change in global surface air temperature would be proportional to sulfate burden. The sulfate forcing $F_S$, the rate of change in Earth heat content due to sulfate, $dH_S/dt$, is proportional to the sulfate burden $S$,

$$F_S = dH_S/dt = -kS$$

where $k$ is the constant of proportionality; the minus sign denotes cooling. One would not necessarily expect temperature change $dT/dt$ to be proportional to $dH_S/dt$, because of time lag and damping. The time lag is only a few years (e.g., Held et al., 2010) so there would seem to be a possibility of discerning change in $T$ resulting from a change in sulfate burden, especially if the forcing is sustained. I think the authors should discuss this.

I also did not pay much attention to Table 2 of the Revision, an addition to the manuscript. It should be explained how the quantities in the table were determined, and the time period to which they pertain. In principle all the quantities – sulfate burden, sulfate wet dep rate, sulfate dry dep rate, sulfate lifetime – are time dependent. What would be much more valuable would be time series of these quantities, also rate coeffs for wet and dry deposition (ratio of dep rate to stock) rather than a simple table. Also sulfate yield. Are those intensive quantities more or less constant with time? Do they change during the time period of interest (1960-1990)? Are these quantities coherent with time series of SAT? Such analyses could turn this paper into an important study.

Held, I.M., Winton, M., Takahashi, K., Delworth, T., Zeng, F. and Vallis, G.K., 2010. Probing the fast and slow components of global warming by returning abruptly to preindustrial forcing. *Journal of Climate*, *23*(9), pp.2418-2427.